# Haematological predictors of poor outcome among COVID-19 patients admitted to an intensive care unit of a tertiary hospital in South Africa

Zivanai C. Chapanduka[1], Ibtisam Abdullah[1], Brian Allwood[2], Coenraad F. Koegelenberg[2], Elvis Irusen[2], Usha Lalla[2], Annalise E. Zemlin[3], Tandi E. Masha[4], Rajiv T. Erasmus[3], Thumeka P. Jalavu[3], Veranyuy D. Ngah[5], Anteneh Yalew[5,6,7], Lovemore N. Sigwadhi[5], Nicola Baines[2], Jacques L. Tamuzi[5], Marli McAllister[5], Anne K. Barasa[8], Valerie K. Magutu[8], Caroline Njeru[8], Angela Amayo[8], Marianne W. Wanjiru Mureithi[8], Mary Mungania[9], Musa Sono-Setati[10], Alimuddin Zumla[11], Peter S. Nyasulu[5,12] *

1 Division of Haematological Pathology, Department of Pathology, Faculty of Medicine and Health Sciences, Stellenbosch University & NHLS Tygerberg Hospital, Cape Town, South Africa, 2 Division of Pulmonology, Department of Medicine, Faculty of Medicine and Health Sciences, Stellenbosch University & Tygerberg Hospital, Cape Town, South Africa, 3 Division of Chemical Pathology, Department of Pathology, Faculty of Medicine and Health Sciences, Stellenbosch University & NHLS Tygerberg Hospital, Cape Town, South Africa, 4 Faculty of Health & Wellness Sciences, Peninsula University of Technology Bellville Campus, Cape Town, South Africa, 5 Division of Epidemiology and Biostatistics, Department of Global Health, Faculty of Medicine and Health Sciences, Stellenbosch University, Cape Town, South Africa, 6 Department of Statistics, College of Natural and Computational Sciences, Addis Ababa University, Addis Ababa, Ethiopia, 7 National Data Management Centre for Health, Ethiopian Public Health Institute, Addis Ababa, Ethiopia, 8 Department of Medical Microbiology, University of Nairobi, Nairobi, Kenya, 9 Department of Laboratory Medicine, Kenyatta National Hospital, Nairobi, Kenya, 10 Department of Public Health Medicine, Department of Health, Limpopo, South Africa, 11 Center for Clinical Microbiology, Division of Infection and Immunity, University College London, NIHR Biomedical Research Centre, UCL Hospitals NHS Foundation Trust, London, United Kingdom, 12 Division of Epidemiology and Biostatistics, School of Public Health, Faculty of Health Sciences, University of the Witwatersrand, Johannesburg, South Africa

* pnyasulu@sun.ac.za

**Data Availability Statement:** All relevant data are within the article and its Supporting Information files.

## Abstract

### Background

Studies from Asia, Europe and the USA indicate that widely available haematological parameters could be used to determine the clinical severity of Coronavirus disease 2019 (COVID-19) and predict management outcome. There is limited data from Africa on their usefulness in patients admitted to Intensive Care Units (ICUs). We performed an evaluation of baseline haematological parameters as prognostic biomarkers in ICU COVID-19 patients.

### Methods

Demographic, clinical and laboratory data were collected prospectively on patients with confirmed COVID-19, admitted to the adult ICU in a tertiary hospital in Cape Town, South Africa, between March 2020 and February 2021. Robust Poisson regression methods and receiver

**Funding:** This work was carried out under the Stellenbosch University Special Vice-Rector (RIPS) Fund and the COVID-19 Africa Rapid Grant Fund supported under the auspices of the Science Granting Councils Initiative in Sub-Saharan Africa (SGCI) and administered by South Africa's National Research Foundation (NRF) in collaboration with Canada's International Development Research Centre (IDRC), the Swedish International Development Cooperation Agency (SIDA), South Africa's Department of Science and Innovation (DSI), the Fonds de Recherche du Québec (FRQ), the United Kingdom's Department of International Development (DFID), United Kingdom Research and Innovation (UKRI) through the Newton Fund, and the SGCI participating councils across 15 countries in sub-Saharan Africa, (grant number PR_COV192109106638722) awarded to PSN. The funders had no role in study design, data collection and analysis, decision to publish, or preparation of the manuscript.

**Competing interests:** The authors have declared that no competing interests exist.

**Abbreviations:** ACE 2, angiotensin-converting enzyme 2; aPTT, activated partial thromboplastin time; ARDS, acute respiratory distress syndrome; AUC, area under curve; CAC, COVID-19 associated coagulopathy; COVID-19, Coronavirus Disease 2019; DIC, Disseminated intravascular coagulation; EDTA, ethylenediaminetetraacetic acid; FBC, full blood count; Hb, haemoglobin; IL, 10 Interleukin 10; ICU, intensive unit care; IL, interleukin; INR, International normalized ratio; MCHC, mean corpuscular haemoglobin concentration; MCV, mean corpuscular volume; MLR, monocyte to lymphocyte ratio; NHLS, National Health Laboratory Service; NLR, neutrophil/lymphocyte; PAI-1, Plasminogen activator inhibitor 1; PCR, Polymerase Chain Reaction; PLR, platelet/lymphocyte ratio; PT, Prothrombin Time; REDCap, Research Electronic Data Capture; RDW, red cell distribution width; ROC, Operating Characteristic Curve; SANAS, South African National Accreditation Services; SARS-COV-2, severe acute respiratory syndrome coronavirus 2; TBH, Tygerberg Hospital; TGF-β, Transforming growth factor β; WBC, white blood cell count.

operating characteristic (ROC) curves were used to explore the association of haematological parameters with COVID-19 severity and mortality.

## Results

A total of 490 patients (median age 54.1 years) were included, of whom 237 (48%) were female. The median duration of ICU stay was 6 days and 309/490 (63%) patients died. Raised neutrophil count and neutrophil/lymphocyte ratio (NLR) were associated with worse outcome. Independent risk factors associated with mortality were age (ARR 1.01, 95%CI 1.0–1.02; p = 0.002); female sex (ARR 1.23, 95%CI 1.05–1.42; p = 0.008) and D-dimer levels (ARR 1.01, 95%CI 1.002–1.03; p = 0.016).

## Conclusions

Our study showed that raised neutrophil count, NLR and D-dimer at the time of ICU admission were associated with higher mortality. Contrary to what has previously been reported, our study revealed females admitted to the ICU had a higher risk of mortality.

## 1. Introduction

The Coronavirus disease 2019 (COVID-19) has caused over 240 million infections and around 5 million deaths globally [1]. South Africa is the worst affected African country with approximately 3 million confirmed cases and 90,000 deaths [1].

Severe Acute Respiratory Syndrome Corona Virus 2 (SARS-CoV-2) is a single-stranded RNA virus that has surface glycoprotein studs on the viral envelope. The angiotensin-converting enzyme (ACE) 2 receptor is the target for SARS-CoV-2 adhesion and is expressed on endothelial cells and multiple organs, including the lung, kidney, and heart [2]. Viral elements have been shown within endothelial cells and this, coupled with the accumulation of inflammatory cells, drives endothelial cell injury. This damage causes endothelial dysfunction with recruitment of immune cells such as mononuclear cells and neutrophils causing further damage and immune activation. Impaired microcirculatory function due to inflammation, thrombosis and endothelial dysfunction could explain the multi-organ dysfunction seen in severe COVID-19 disease [3]. The lympho-haematopoietic and immune systems are affected to a variable degree, in SARS-CoV-2 infection.

Laboratory investigations play an important role in the diagnosis and treatment of COVID-19. Dysregulation of the immune system with a massive increase in cytokine production, termed the cytokine storm, is associated with haematological changes in the full blood count (FBC) parameters, morphological features of blood cells and haemostatic parameters [4, 5]. Haemostasis-related abnormalities seen in COVID-19 are referred to as COVID-19 associated coagulopathy (CAC). Leucocytosis, neutrophilia, lymphopenia, increased neutrophil/lymphocyte ratio (NLR), monocytosis, anaemia, thrombocytopenia, increased platelet/lymphocyte ratio (PLR) and thrombocytosis can be detected on FBC and blood film examination [6, 7]. The thrombocytopenia could be due to direct bone marrow suppression, destruction of platelets by autoantibodies and consumption of platelets as a result of endothelial injury, lung injury and the thrombosis [8]. Increased PLR and NLR at the time of admission has been suggested to predict the severity and poor prognosis in COVID-19 [9, 10].

CAC varies from mild derangement of laboratory tests of haemostasis without any clinical abnormality, to sepsis-induced coagulopathy and possibly, overt disseminated intravascular coagulation. Both inflammatory cells and haemostatic components orchestrate CAC which is characterised by a prominent thrombotic component with reduction of coagulation factors and limited activation of fibrinolysis. Immunothrombosis is likely to start with endothelial cell injury by viral invasion through the ACE2 receptor which results in endotheliitis with an infiltrate of inflammatory cells and disruption of the natural antithrombotic state. The endotheliitis, inflammatory cytokine storm, complement system (by activating coagulation factors) and hypoxia synergistically activate each other resulting in hypercoagulability [11]. Additionally, CAC is associated with hypofibrinolysis within the alveolar space due to increased local production of Plasminogen activator inhibitor 1(PAI-1) [12].

While the main reported clinical manifestation of CAC is thrombosis, the changes in haemostatic parameters in COVID-19 include elevation of D-dimer, raised fibrinogen, prolonged prothrombin time (PT), activated partial thromboplastin time (aPTT) as well as the presence of lupus anticoagulants [6, 13, 14]. Researchers suggested that the source of D-dimer is in not only the activation of the intravascular haemostatic system but also activation of bronchoalveolar haemostatic system as a part of the host defence mechanisms [15, 16]. Patients with COVID-19 infection are categorised based on clinical symptoms and laboratory results into a mild, moderate, or severe infection [17]. The severity of clinical presentations has been associated with high viral load, increasing age, male sex, and the presence of comorbidities [18], resulting in higher risks of acute respiratory distress syndrome (ARDS), organ failure and death [19]. Most COVID-19 patients have mild to moderate symptoms and recover. Conversely, the mortality rate is as high as 60% in patients admitted to intensive care unit (ICU) due to severe COVID-19 infection rendering the early identification and management of these cases, crucial [20]. Several reports from Asia, Europe, and the United State of America (USA) indicate that derangement of haematological parameters may be associated with a poor prognosis [7, 13, 21, 22]. However, the evidence so far appears inconsistent.

South Africa has a socioeconomically diverse population and several endemic diseases. This adds an extra layer of complexity and limits the local applicability of the available international data. Furthermore, in Sub-Saharan Africa, the data on haematological parameters, in general and specifically in an ICU setting, is scanty.

The aim of this study was to evaluate baseline "routine" haematological parameters in patients admitted to the ICU of a tertiary hospital in the Western Cape province of South Africa during the first and second COVID-19 waves and correlate these to the disease severity and outcome.

## 2. Methods

### 2.1. Study design

This prospective cohort study was conducted at Tygerberg Hospital (TBH) during the first two waves of the COVID-19 pandemic between 27 March 2020 and 10 February 2021. The TBH is a 1380-bed hospital that serves as the main teaching hospital for Stellenbosch University Faculty of Medicine and Health Sciences and was designated as a centre for COVID-19 management with additional critical care service. It provides tertiary service to around 3.5 million people.

### 2.2. Sample size

The study included data of 490 patients admitted to the adult ICU with a positive SARS-CoV-2 polymerase chain reaction (PCR) during the above-mentioned dates. Details regarding

admission criteria are documented in the Western Cape Government's provincial guidelines [23]. Furthermore, patient admission depended on the availability of ICU beds.

## 2.3. Data collection

Clinical data were extracted from ICU clinical notes and entered onto a REDCap® (Research Electronic Data Capture) database a secure web application for building. Haematological parameters were imported from the National Health Laboratory Service (NHLS) Laboratory Information System (TrakCare® Lab Enterprise) onto the database. Data quality assessment was undertaken by the research coordinator and verified by the research supervisor to ensure data quality, reliability and to maintain method consistency.

## 2.4. Ethics

This study was approved by the Health Research Ethics Committee of Stellenbosch University, approval number: N20/04/002_COVID-19. Patient confidentiality was ensured by labelling data with a unique episode number. The research project followed the laid down guidelines in the ethical conduct of studies involving human participants [24].

## 2.5. Laboratory analyses

Blood samples for FBC and coagulation parameters were collected in ethylenediaminetetraacetic acid (EDTA) and buffered sodium citrate tubes respectively, on ICU admission of all study participants. FBC samples were analysed on the Siemens ADVIA 2120i™ (Siemens AG Healthcare, 91052 Erlangen, Germany) and coagulation tests were performed on the Sysmex Cs-2000i™ coagulation analyser (Sysmex Medical Electronics, Kobe, Japan). The NHLS Haematology Department is accredited by the South African National Accreditation System (SANAS) and participates in various internal and external quality assurance programmes to meet ISO 15189 standards. The following haematological parameters were determined: white blood cell count (WBC), haemoglobin (Hb), mean corpuscular volume (MCV), mean corpuscular haemoglobin concentration (MCHC), red cell distribution width (RDW), platelet, neutrophil, lymphocyte, monocyte, eosinophil and basophil counts, D-dimer, PT, aPTT and fibrinogen.

## 2.6. Outcomes and predictors variables

Data collected included sociodemographic (age, sex, smoking status), pre-existing comorbidities associated with severe COVID-19 outcome (hypertension, diabetes mellitus and hyperlipidaemia) and haematological parameters. The NLR, PLR and monocyte to lymphocyte ratio (MLR) were calculated. The primary outcome was the proportion of patients who died (nonsurvivors) after admission to the ICU. Time to death or discharge and length of stay in ICU were also evaluated.

## 2.7. Statistical analysis

Continuous variables were expressed as median with inter-quartile range since they were nonnormal data. Categorical variables were expressed using frequencies and percentages. Chisquared and Fisher exact test were used to assess the association between mortality and the categorical variables. Median test was used to assess the equality of the median of the continuous variables between non-survivor and survivor groups. Robust Poisson regression was used to assess significant association between demographic, laboratory results, and mortality. Factors associated with death at $p < 0.15$ in unadjusted univariable robust Poisson regression were

included in a multivariable model to identify predictor variables associated with death. Due to the high mortality, around 63%, the logistic regression overestimated the effect measure with large standard errors resulting in wide confidence intervals, therefore robust Poisson regression was used. Adjusted risk ratios and their 95% CIs were used as a measure of association. Receiver Operating Characteristic curve (ROC) analysis was performed to evaluate the diagnostic performance of various haematological parameters to discriminate between severe cases in terms of survival and non-survival. Factors with $p < 0.05$ were considered significantly associated with mortality. All statistical analyses were performed using Stata (V.16, Stata Corp, College Station, Texas, USA) and R (V, 4.1.0, R Core Team) with R Studio (V.1.4.1, R Studio Team) statistical software.

## 3 Results

### 3.1. Patient demographics

In this cohort, 490 patients were admitted to the ICU from March 2020 to February 2021. The median age was 54.1 years (IQR: 46.1–61.3). Those at a higher risk of death had a significantly higher median age (56.3 vs 50.0 years, $p < 0.001$) than those discharged (Table 1). More males than females were admitted during this period; 52% vs 48%. The proportion of females who died was significantly higher than males; 53% vs 41%, p = 0.011. Further analysis of the data, to elucidate the association of gender and comorbidities with mortality did not reveal any association between mortality and comorbid conditions namely, obesity, hypertension, diabetes mellitus and acute kidney injury (S1 Table). Age was significantly higher among non-survival females ($p < 0.001$) and further analysis was carried out to identify any gender differences between the various age categories. This analysis demonstrated no significant gender difference between the various age categories, as summarised in S2 Table.

The prevalence of the underlying comorbidities considered in this analysis was hypertension,59%, diabetes mellitus,50%, hyperlipidaemia,10% and smoking, 22%.

Sixty-three percent (309/490) of the patients admitted in ICU died and the median duration of ICU stay was 6 days (IQR: 3–10). Median duration of ICU stay differed significantly in those who died, who stayed 2 days longer (7 vs 5 days, p = 0.001). Table 1 summarises patient demographics, co-morbidities, smoking history and outcomes.

### 3.2. Baseline haematological parameters

The association between baseline haematological parameters among survivors and non-survivors is shown in Table 2. The median MCHC was 0.25 pg higher in non-survivor females (p = 0.018) than in those who were discharged from ICU. There was a statistically significant difference in neutrophil count (11.04 vs 9.70, p = 0.03); as well as a higher NLR (10.1 vs 8.4,

**Table 1. Sociodemographic and risk factors of COVID-19 patients admitted to the ICU.**

| Characteristic | participants | Combined | Discharge | Death | p-value |
|---|---|---|---|---|---|
| Participants | | 490 | 181 | 309 | |
| Age (in years) | 490 | 54.1 (46.1–61.3) | 50.0 (42.7–58.3) | 56.3 (47.8–62.3) | <0.001 |
| Gender (Female) | 490 | 237 (48%) | 74 (41%) | 163 (53%) | 0.011 |
| Smoker | 273 | 59 (22%) | 23 (21%) | 36 (22) | 0.867 |
| Duration of ICU stay | 490 | 6 (3–10) | 5 (3–10) | 7 (3–10) | 0.001 |
| Hypertension | 466 | 276 (59%) | 94 (54%) | 182 (62%) | 0.099 |
| Diabetes mellitus | 467 | 233 (50%) | 78 (45%) | 155 (53%) | 0.111 |
| Hyperlipidaemia | 466 | 48 (10%) | 14 (8%) | 34 (12%) | 0.228 |

**Table 2. Baseline haematological characteristics of COVID-19 patients admitted to the ICU.**

| Characteristics | Total participants | Reference interval | combined | Discharge | Death | p-value |
|---|---|---|---|---|---|---|
| Number | | | 482 | 181 | 301 | |
| WCC | 482 | 3.92–10.40 x 10$^9$/L | 11.3 (8.3–14.6) | 10.9 (7.8–13.4) | 11.6 (8.9–15.3) | 0.110 |
| RCC | 482 | 3.80–4.80 x10$^{12}$/L | 4.5 (4.1–4.9) | 4.5 (4.2–4.9) | 4.5 (4.0–4.9) | 0.836 |
| Hb | 252 | Male:13.0–17.0g/dL; | 13.6 (12.6–14.6) | 13.5 (12.5–14.5) | 13.7 (12.7–14.7) | 0.332 |
| | 230 | Female:12.0–15.0g/dL | 12.5 (11.3–13.3) | 12.75 (12.1–13.6) | 12.2 (11.1–13.25) | 0.111 |
| MCV | 252 | Male:83.1–101.6 femtoliters | 89.15 (84.9–93) | 88.7 (83.8–92.8) | 89.8 (85.2–93) | 0.251 |
| | 230 | Female: 78.9–98.5 femtoliters | 88.55 (84.5–92.1) | 88.15 (84.1–92.3) | 88.65 (84.55–92.1) | 0.778 |
| MCHC | 226 | Male 33.0–35.0 g/dL | 32.8 (31.8–33.7) | 32.6 (31.6–33.6) | 32.8 (31.9–33.7) | 0.725 |
| | 180 | Female: 32.7–34.9 g/dL | 32.4 (31.15–33.4) | 32.95 (31.9–33.7) | 33.2 (31–33.3) | 0.018 |
| RDW | 226 | Male:12.1–16.3% | 13.6 (13.1–14.3) | 13.6 (13.2–14.4) | 13.7 (13–14.3) | 0.412 |
| | 180 | Female:12.4–17.3% | 14.1 (13.4–14.65) | 13.9 (13.4–14.5) | 14.1 (13.4–14.9) | 0.434 |
| Platelets | 482 | 186-454x 10$^9$/L | 297.5 (225–376) | 304 (217–384) | 288 (230–370) | 0.397 |
| Neutrophils | 480 | 1.60–6.98x 10$^9$/L | 10.31 (7.37–15.59) | 9.70 (6.65–15.23) | 11.04 (7.64–15.93) | 0.030 |
| Lymphocytes | 479 | 1.40–4.20x 10$^9$/L | 1.04 (0.67–1.70) | 1.06 (0.69–1.68) | 1.04 (0.67–1.72) | 0.897 |
| Monocytes | 479 | 0.30–0.80x 10$^9$/L | 0.50 (0.32–0.85) | 0.47 (0.31–0.80) | 0.53 (0.33–0.88) | 0.217 |
| Eosinophils | 74 | 0.00–0.95x 10$^9$/L | 0.1(0.1–0.3) | 0.1 (0.1–0.3) | 0.1 (0.1–0.3) | 0.948 |
| Basophils | 478 | 0.00–0.10 x 10$^9$/L | 0.05 (0.02–0.10) | 0.05 (0.03–0.12) | 0.04 (0.02–0.10) | 0.146 |
| D-dimer | 464 | 0.00–0.25mg/L | 1.06 (0.45–4.24) | 0.62 (0.36–2.32) | 1.33 (0.53–5.97) | <0.001 |
| INR | 413 | 11–13.5 seconds | 1.13 (1.05–1.22) | 1.11 (1.04–1.20) | 1.14 (1.06–1.24) | 0.083 |
| PTT | 134 | 22–30.7 seconds | 24.6 (20.2–28.7) | 24.85 (18.8–28.15) | 24.4 (20.7–28.9) | 0.904 |
| Fibrinogen | 6 | 2-4g/L | 7.3 (6–8.6) | 5.05 (3.9–6.2) | 8.5 (7.2–8.8) | 0.083 |
| Platelets / Lymphocytes ratio | 479 | | 267.7 (153.8–425.5) | 270.0 (151.1–429.4) | 261.2 (154.9–414.3) | 0.750 |
| Neutrophils / Lymphocytes ratio | 479 | | 9.4 (6.2–16.1) | 8.4 (5.7–14.2) | 10.1 (6.8–17.4) | 0.012 |
| Monocytes / Lymphocytes ratio | 479 | | 0.48 (0.31–0.69) | 0.45 (0.30–0.64) | 0.49 (0.32–0.73) | 0.263 |

Abbreviations: Hb: Haemoglobin; INR: international normalized ratio; MCHC: mean corpuscular haemoglobin concentration; PTT: partial thromboplastin time; RCC: Red Cell Count; RDW; red cell distribution width; WCC: White cell count

p = 0.012) among non-survivors. The median D-dimer in non-survivors was significantly higher than in those who survived (1.33 mg/L vs 0.62 mg/L, p<0.001).

### 3.3. Association of socio-demographic and haematologic parameters with mortality status

Table 3 presents data that shows the association between socio-demographic and haematologic parameters with mortality as an outcome in our cohort. In the multivariable analysis, age, female sex, and D-dimer, were independent risk factors for mortality. With a RR 1.01 (p = 0.002), for every one-year increase in age, the risk of death increases by 1%. For female sex, the RR was 1.23 (1.05–1.42), p = 0.008, showing the risk of death was 23% higher among females, compared to males. The D-dimer, RR 1.01 (1.002–1.03) p = 0.016, showed that for every mg/L increase in D-dimer level, the risk of death increases by 2%.

### 3.4. Kaplan-Meier survival estimates between males and females admitted to the ICU (Fig 1)

The rate of death was higher among female patients during the first week of admission. This rate was similar in the two weeks follow-on period. However, females who survived the first

**Table 3. Association of socio-demographic and haematology parameters with mortality status in COVID-19 ICU patients.**

| Characteristic | Unadjusted RR (95% CI) | p-value | Adjusted RR (95% CI) | p-value |
|---|---|---|---|---|
| Age | 1.01 (1.01–1.02) | <0.001 | 1.01 (1.0–1.02) | 0.002 |
| Sex: Female | 1.19 (1.04–1.37) | 0.012 | 1.23 (1.05–1.42) | 0.008 |
| Duration in ICU stay | 1.00 (0.99–1.01) | 0.809 | | |
| Hypertension | 1.13 (0.97–1.31) | 0.107 | 0.93 (0.79–0.1.09) | 0.370 |
| Diabetes Mellitus | 1.12 (0.97–1.29) | 0.113 | 1.13 (0.96–1.31) | 0.136 |
| Hyperlipidaemia | 1.14 (0.93–1.39) | 0.182 | | |
| WCC | 1.00 (0.99–1.01) | 0.559 | | |
| RCC | 0.98 (0.87–1.10) | 0.703 | | |
| Hb | 0.98 (0.94–1.01) | 0.218 | | |
| MCV | 1.00 (0.99–1.01) | 0.745 | | |
| MCHC | 0.96 (0.92–1.02) | 0.167 | | |
| RCDW | 1.00 (0.94–1.06) | 0.928 | | |
| Platelets | 1.00 (1.00–1.00) | 0.301 | | |
| Neutrophils | 1.00 (0.998–1.003) | 0.600 | | |
| Lymphocytes | 0.99 (0.98–1.01) | 0.576 | | |
| Monocytes | 0.99 (0.96–1.03) | 0.704 | | |
| Eosinophils | 0.83 (0.41–1.69) | 0.607 | | |
| Basophils | 0.84 (0.52–1.36) | 0.472 | | |
| D-dimer | 1.02 (1.01–1.03) | <0.001 | 1.01 (1.002–1.03) | 0.016 |
| INR | 1.15 (1.00–1.33) | 0.046 | 1.07 (0.79–1.45) | 0.667 |
| PTT | 1.00 (0.99–1.01) | 0.789 | | |
| Fibrinogen | 1.46 (0.91–2.37) | 0.119 | | |
| Platelets / Lymphocytes ratio | 1.00 (0.9997–1.0002) | 0.884 | | |
| Neutrophils / Lymphocytes ratio | 1.01 (1.002–1.013) | 0.007 | 1.00 (0.99–1.01) | 0.948 |
| Monocytes / Lymphocytes ratio | 1.21 (1.07–1.38) | 0.003 | 1.14 (0.95–1.37) | 0.165 |

Abbreviations: Hb: Haemoglobin; INR: international normalized ratio; MCHC: mean corpuscular haemoglobin concentration; PTT: partial thromboplastin time; RCC: Red Cell Count; RDW; red cell distribution width; WCC: White cell count

week of ICU admission seem to have a longer survival duration than males. The length of stay in the ICU was $\geq$ 60 days for females compared to 45 days for males in this cohort

### 3.5 ROC curves (Figs 2 and 3)

The ROC curves for D-dimer and international normalized ratio (INR) were mostly non-significant, with the area under curve (AUC) approximating 0.5. The inference is that these tests do not discriminate between survivors and non-survivors. The optimal D-dimer cut-off value at admission for predicting mortality is $\geq$ 0.740 mg/L (sensitivity of 67.4% and a specificity of 57.2%) as shown in Table 4. In addition, MLR had a good specificity of 92% but a very low sensitivity with a cut-off value of 1.12 for predicting mortality.

## 4. Discussion

Although several studies have evaluated haematological parameters in COVID-19, there are scanty data from South Africa and other African countries. This ICU-based study compared widely available routine haematological parameters in COVID-19 South African patients in ICU who died (non-survivors) and those who were discharged (survivors), with the aim of

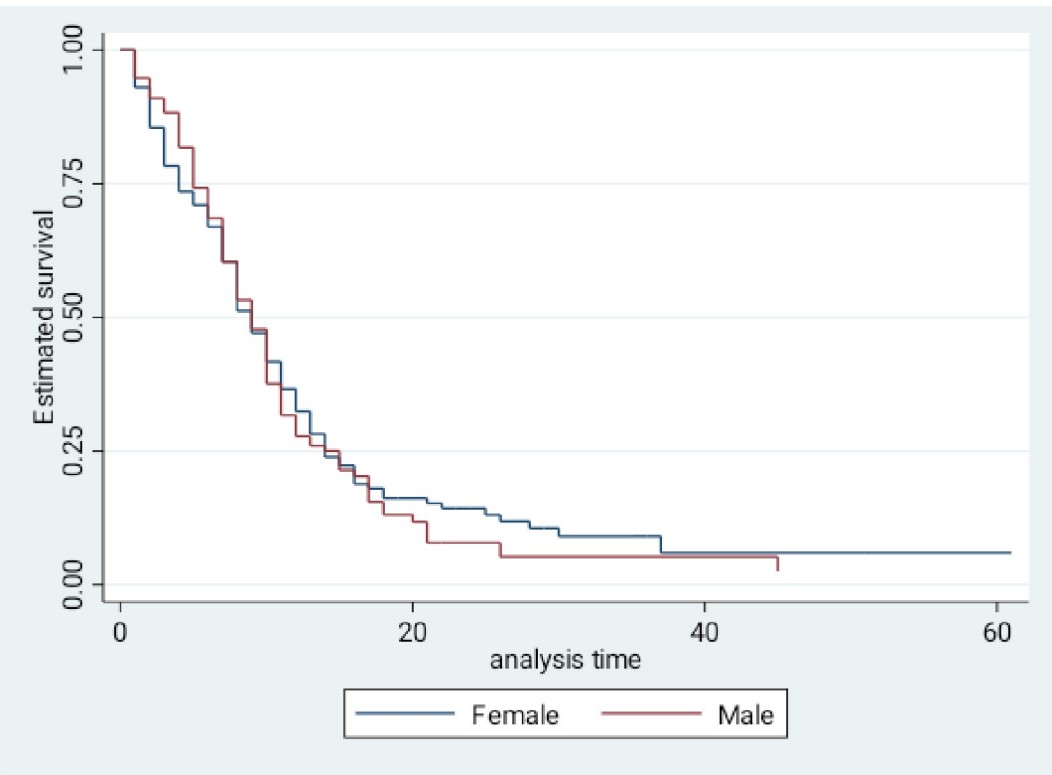

**Fig 1. Kaplan-Meier product limit estimates of survival among COVID-19 patients admitted to the ICU.**

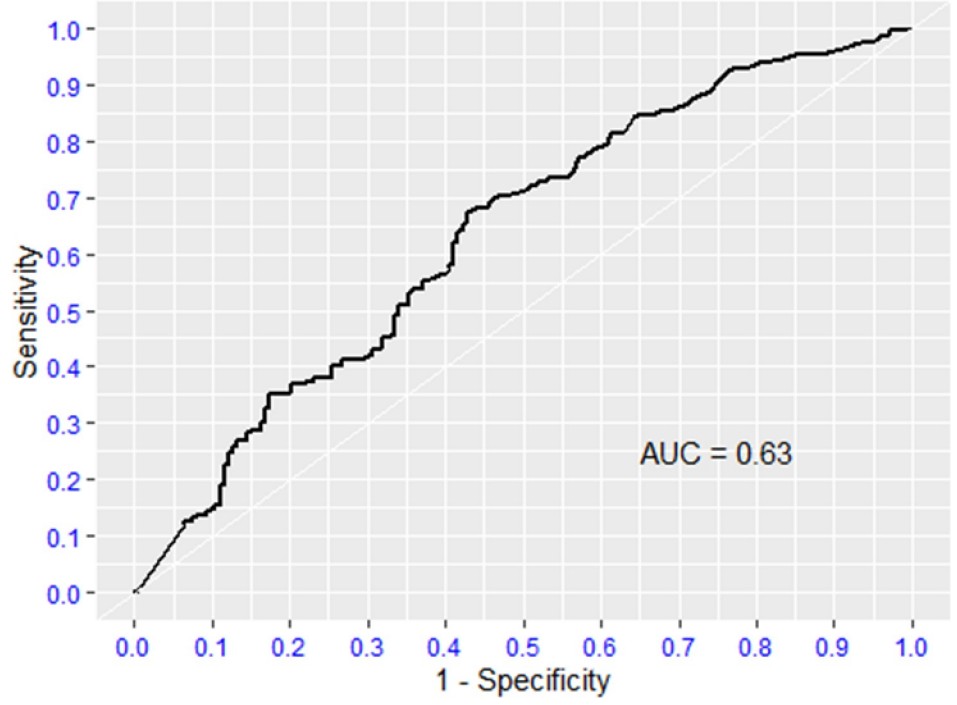

**Fig 2. ROC curve D-dimer.**

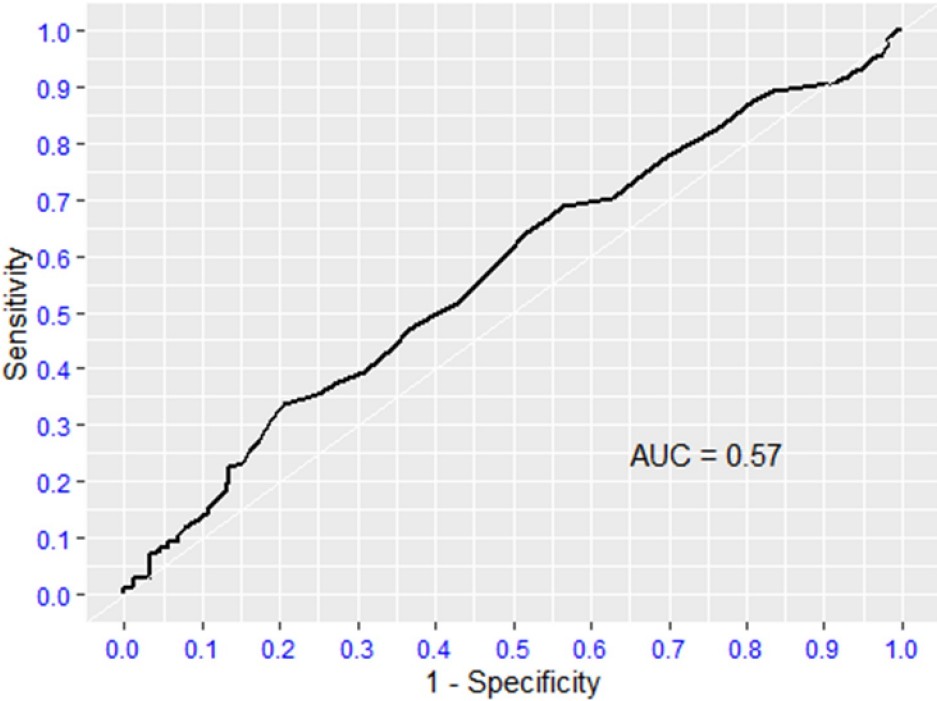

**Fig 3. ROC curve INR.**

establishing which of the haematologic parameters, comorbidities, and demographic attributes, affected the patient outcomes.

The study found that raised neutrophil count, NLR, D-dimer and MCHC levels were associated with worse outcomes in this ICU cohort. A multivariable analysis showed that elevated D-dimer level was an independent risk factor for mortality.

Interestingly, we found a significantly higher MCHC in non-survivor females, compared to survivors. This is like the findings of Hesse *et al* who reported that elevated MCHC is associated with worse outcome [25]. However, there was no difference in the RDW, MCV and haemoglobin values between survivors and non-survivors regardless of analysis by sex. Thus, we could not confirm the findings of Pouladzadeh *et al* whose study concluded that RDW may serve as an independent predictor of the severe infection and mortality in COVID-19 patients [26]. In a study of 143 ICU patients, a high RDW above 13% was shown to be a predictor of COVID-19 related mortality [27].

In our study, the total leucocyte count had no impact on the patient outcomes even though previous studies demonstrated increased mortality in COVID-19 patients with a higher leucocyte count [21, 28, 29]. Non-survivors had a higher neutrophil count compared to the survivor group, although the difference was not statistically significant. Importantly, our analysis

**Table 4. ROC analysis of promising biomarkers for severity of COVID-19.**

| Predictors | Direction | Optimal cut point | Sensitivity | Specificity | AUC |
|---|---|---|---|---|---|
| D-Dimer | > = | 0.74 | 0.67 | 0.57 | 0.63 |
| INR | > = | 1.21 | 0.34 | 0.79 | 0.57 |
| MLR | < = | 1.12 | 0.17 | 0.93 | 0.54 |
| NLR | > = | 8.73 | 0.60 | 0.57 | 0.58 |

revealed a significantly higher NLR in the non-survivor group compared the survivor group. The NLR is a reliable biomarker for the evaluation of systemic inflammatory response and is recommended as a predictor of COVID-19 severity and mortality [10]. Liu *et al.*, 2020, investigated the baseline NLR of 245 COVID-19 patients admitted to hospital and found 8% higher risk of in-hospital mortality for each unit increase in NLR and 15-fold higher risk of death in patients with high NLR tertile [30]. However, in addition to high NLR, these patients were found to be older, more likely to be males and have underlying comorbidities, compared with subjects in the lowest tertile of NLR [30]. Núñez *et al.*, 2020, investigated baseline characteristics of 282 hospitalised COVID-19 patients and demonstrated that high NLR, at the moment of hospitalisation, predicts poor outcomes including death or critical care admission [31]. In contrast to our findings, they reported that lymphopenia is a good independent predictor for ICU admission, mechanical ventilation, and death [31]. In our ICU cohort, the increased NLR was mostly due to the increase in neutrophils and lymphopenia was a constant f, regardless of the outcome. Unlike our study which focuses on ICU patients with severe COVID-19, previous studies investigated the baseline NLR for patients at hospitalisation with variable COVID-19 severity [30, 31]. On the other hand, Ma *et al.*, 2020, investigated baseline characteristics for 81 patients with severe COVID-19 and found that those with a high NLR have a higher incidence of ARDS and a higher rate of mechanical ventilation [32]. Comparably, analysis of leucocyte and neutrophil counts and the NLR in our ICU cohort, confirms the superiority of the NLR in the severe COVID-19 setting.

The MLR is recommended by some studies to predict disease outcome [33]. A study that compared host immunity between survivor and non-survivor COVID-19 patients in Wuhan, China, showed a selective reduction of antigen-presenting monocytes in non-survivor group which could be due to expression of immunosuppressive cytokines, such as interleukin-10 (IL-10), transforming growth factor-β (TGF-β), resulting in immunosuppression [34, 35]. Another study reported an elevated monocyte count in patients with early-stage COVID-19 while a lower monocyte count was reported in ICU patients [21, 36]. Neither the monocyte counts nor the MLR was found to have any correlation with patient outcomes in our cohort. Although, the MLR was higher in the non-survivor group, the difference was not statistically significant. Interestingly, a local study during the South African first wave compared laboratory parameters in COVID-19 patients between critical and non-critical patients and found no significant difference in monocyte count between critical and non-critical COVID-19 patients [25].

D-dimer levels, a marker that is mostly known to be associated with thrombosis, has been by far the most reliable and consistent predictor of mortality. A study that included 343 COVID-19 patients from Wuhan, China measured D-dimer levels at hospital admission. The study concluded that D-dimer levels $\geq$ 2.0 µg/mL (fourfold increase) is a predictor of mortality with a sensitivity of 92.3% and specificity of 83.3% [37], while another study that analysed 1114 admitted COVID-19 patients, found D-dimer value of 2.025mg/L to be the optimal probability cut-off for prognosis of death. D-dimer levels were significantly higher in patients who died than in survivors. Higher D-dimer levels were observed in patients with advanced age, male gender, and some underlying diseases (hypertension, coronary heart disease, cerebrovascular disease, renal insufficiency) which are factors associated with worse outcome [38]. Two meta-analyses, one including 10 studies with 1430 (378 severe and 1052 non-severe) COVID-19 patients and the second including 6 studies with 1329 COVID-19 patients, all from China, concluded that patients with worse clinical outcome (ICU admission and mortality) have significantly higher D-dimer levels compared to non-severe patients. Like other studies, a meta-analysis by Du WN *et al* found that age was a confounding factor in the poor clinical outcome group and that disease severity was significantly associated with age in patients >55 years. The authors highlighted the high heterogeneity of studies and the lack of external validation as potential limitations [39,

40]. Short *et al* aimed to overcome some of other studies' limitations by adjusting for confounder factors of mortality such as presence of comorbidities through multivariate logistic regression analysis. Additionally, they had a large sample size with 3418 ICU admitted patients from 86 hospitals in the United States. The D-dimer was measured within the first 2 days following ICU admission and the outcome was measured on day 28. The investigators confirmed that higher D-dimer levels were independently associated with an increased risk of death in COVID-19 patients, even after adjustment for many confounders [41]. However, a clear cut-off level was not provided given the variability of the D-dimer testing methods. Similarly, Poudel *et al* performed a retrospective study which included 4 tertiary levels hospitals in Nepal [42]. They assessed the D-dimer levels of COVID-19 patients at admission and concluded that the level of D-dimer at admission is an accurate marker for predicting patient mortality [42]. They proposed that a D-dimer level of 1.5 μg/ml at admission is the optimal cut-off value for predicting mortality (sensitivity of 70.6% and a specificity of 78.4%) [42]. Although these 4 hospitals have the same reference ranges, they used different kits to measure D-dimer which in addition to a selection bias is a major limitation of their study [42].

Although the above-mentioned studies confirm the value of D-dimer in predicting mortality, the variation in methods and units of measurement of D-dimer limits the worldwide applicability of these studies and highlights the need for on-site analysis and determination of local D-dimer cut-offs. In our study, the D-dimer levels were significantly higher in the non-survivor group, compared to the survivor group. A D-dimer level greater or equal to 0.740 mg/L at admission was found to be the optimal D-dimer cut-off value for predicting mortality in our study (sensitivity;67.4%, specificity;57.2%). The INR, PTT, and fibrinogen, on the other hand, showed no difference between the survivor and non-survivor groups.

Interestingly, the early initiation of anticoagulants did not reduce the risk of death in patients with high D-dimer levels. This supports the position that the elevated D-dimer levels do not necessarily signify intravascular coagulation but may reflect activation of the bronchoalveolar haemostatic system [43]. Local thrombi in the lungs may represent a positive host response and may not require treatment in the absence of respiratory compromise. Uncontrolled thrombo-inflammation may disseminate into the systemic circulation. This would most likely benefit from the use of anticoagulants. Further research is required to look for biomarkers to determine at which stage this beneficial localised thrombo-inflammatory effect changes to become harmful or systemic.

The platelet count did not show a significant association with disease outcome. Several studies suggested the use of PLR as a novel prognostic marker to predict COVID-19 severity [7]. A meta-analysis that included 7 studies with 998 patients, mainly from China, concluded an association between high PLR at admission and disease severity [9]. Our study did not reveal a significant association between PLR and disease outcome. This could be due to the study population, as we included only severe COVID-19 patients while most studies compared PLR between severe and non-severe patients. A nationwide South African study during COVID-19 first wave also failed to find significant difference in PLR in severe and non-severe COVID-19 patients [25].

In contrast to previous studies that demonstrated higher mortality among males, our study revealed a higher mortality rate among female patients [44, 45]. The number of male and female patients in the study population was balanced. We interrogated the role of age and comorbidities in the observed worse outcomes among female patients and found no association. The comorbidities specifically analysed were obesity, hypertension, diabetes and acute kidney injury. Advanced age had a statistically significant correlation with the risk of mortality in the study population regardless of gender (Tables 1 and S2). Considering that the Western

Cape Province of South Africa has a unique ethnic profile, other possible risk factors such as ethnicity and host genetic predisposition need to be explored further [46].

The strengths of our study include the fact that all the patients were admitted to the same ICU and that all samples were analysed on admission in the same laboratory ensuring harmonisation of the pre-analytical and analytical phases of testing. Our study had some limitations; we had a small patient population and only analysed baseline ICU admission laboratory data. A larger sample population may have increased the statistical significance of our markers. Ideally, we would have analysed the trend over their ICU stay, but the numbers were too small on subsequent days.

## 5. Conclusion

This study showed that raised neutrophil count, NLR and D-dimer at the time of ICU admission were associated with higher mortality. The optimal D-dimer cut-off value for predicting mortality is $\geq$ 0.740 mg/L. Surprisingly, our study revealed females admitted to the ICU have a higher risk of mortality. Early identification of patients at high risk of poor prognosis and in whom earlier interventions may be effective in reducing COVID-19 mortality in the ICU, is recommended. Further larger studies are needed to evaluate the use of these biomarkers to guide treatment decisions.

## Supporting information

**S1 Table. Multivariable analysis assessing the association between gender, comorbidities on mortality among the COVID-19 patients admitted in ICU.**
(DOCX)

**S2 Table. Stratified analysis of gender by different age groups among the COVID-19 patients admitted in ICU.**
(DOCX)

## Acknowledgments

Sir Prof Alimuddin Zumla, is co-Principal Investigator of the (PANDORA-ID-NET), the Pan-African Network for Rapid Research, Response, Relief and Preparedness for Infectious Disease Epidemics, supported by the EDCTP. He is in receipt of a UK National Institutes of Health Research, Senior Investigator Award and is a Mahathir Foundation Science Award laureate. We would also like to acknowledge the Stellenbosch University COVID-19 Rapid Research Response Collaboration.

## Author Contributions

**Conceptualization:** Zivanai C. Chapanduka, Ibtisam Abdullah, Jacques L. Tamuzi, Peter S. Nyasulu.

**Data curation:** Zivanai C. Chapanduka, Brian Allwood, Elvis Irusen, Usha Lalla, Thumeka P. Jalavu, Veranyuy D. Ngah, Anteneh Yalew, Lovemore N. Sigwadhi, Nicola Baines, Marli McAllister.

**Formal analysis:** Anteneh Yalew, Lovemore N. Sigwadhi.

**Funding acquisition:** Anne K. Barasa, Peter S. Nyasulu.

**Investigation:** Zivanai C. Chapanduka, Brian Allwood, Usha Lalla, Tandi E. Masha, Rajiv T. Erasmus, Thumeka P. Jalavu, Veranyuy D. Ngah, Jacques L. Tamuzi, Marli McAllister,

Anne K. Barasa, Caroline Njeru, Angela Amayo, Marianne W. Wanjiru Mureithi, Mary Mungania, Musa Sono-Setati, Alimuddin Zumla.

**Methodology:** Zivanai C. Chapanduka, Ibtisam Abdullah, Coenraad F. Koegelenberg, Annalise E. Zemlin, Anteneh Yalew, Lovemore N. Sigwadhi, Jacques L. Tamuzi, Marli McAllister, Valerie K. Magutu, Angela Amayo, Mary Mungania, Peter S. Nyasulu.

**Project administration:** Brian Allwood, Annalise E. Zemlin, Rajiv T. Erasmus, Veranyuy D. Ngah, Nicola Baines, Anne K. Barasa, Valerie K. Magutu, Angela Amayo, Marianne W. Wanjiru Mureithi, Peter S. Nyasulu.

**Resources:** Elvis Irusen, Tandi E. Masha, Rajiv T. Erasmus, Thumeka P. Jalavu, Anne K. Barasa, Caroline Njeru, Musa Sono-Setati, Peter S. Nyasulu.

**Software:** Lovemore N. Sigwadhi, Jacques L. Tamuzi, Caroline Njeru.

**Supervision:** Brian Allwood, Annalise E. Zemlin, Tandi E. Masha, Nicola Baines, Anne K. Barasa, Marianne W. Wanjiru Mureithi, Mary Mungania, Peter S. Nyasulu.

**Validation:** Coenraad F. Koegelenberg, Tandi E. Masha, Veranyuy D. Ngah, Nicola Baines, Jacques L. Tamuzi, Marli McAllister.

**Writing – original draft:** Zivanai C. Chapanduka, Ibtisam Abdullah, Brian Allwood, Coenraad F. Koegelenberg, Elvis Irusen, Annalise E. Zemlin, Thumeka P. Jalavu, Veranyuy D. Ngah, Anteneh Yalew, Lovemore N. Sigwadhi, Jacques L. Tamuzi, Anne K. Barasa, Valerie K. Magutu, Alimuddin Zumla, Peter S. Nyasulu.

**Writing – review & editing:** Zivanai C. Chapanduka, Ibtisam Abdullah, Brian Allwood, Coenraad F. Koegelenberg, Elvis Irusen, Usha Lalla, Annalise E. Zemlin, Tandi E. Masha, Rajiv T. Erasmus, Thumeka P. Jalavu, Veranyuy D. Ngah, Anteneh Yalew, Lovemore N. Sigwadhi, Nicola Baines, Jacques L. Tamuzi, Marli McAllister, Anne K. Barasa, Valerie K. Magutu, Caroline Njeru, Angela Amayo, Marianne W. Wanjiru Mureithi, Mary Mungania, Musa Sono-Setati, Alimuddin Zumla, Peter S. Nyasulu.

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
