## [Decision Letter · Decision Letter 0]

29 Nov 2021

PONE-D-21-33196Hematological predictors of poor outcome among COVID-19 patients admitted to an intensive care unit of a tertiary hospital in South AfricaPLOS ONE

Dear Dr. Nyasulu,

Thank you for submitting your manuscript to PLOS ONE. After careful consideration, we feel that it has merit but does not fully meet PLOS ONE’s publication criteria as it currently stands. Therefore, we invite you to submit a revised version of the manuscript that addresses the points raised during the review process.

Please submit your revised manuscript within 60 days of receipt of this email. If you will need more time than this to complete your revisions, please reply to this message or contact the journal office at plosone@plos.org. Please include the following items when submitting your revised manuscript:A rebuttal letter that responds to each point raised by the academic editor and reviewer(s). You should upload this letter as a separate file labeled 'Response to Reviewers'.A marked-up copy of your manuscript that highlights changes made to the original version. You should upload this as a separate file labeled 'Revised Manuscript with Track Changes'.An unmarked version of your revised paper without tracked changes. You should upload this as a separate file labeled 'Manuscript'.

We look forward to receiving your revised manuscript.

Kind regards,

Marlene Camacho-Rivera, ScD, MPH

Academic Editor

PLOS ONE

Journal Requirements:

"The funders had no role in study design, data collection and analysis, decision to publish, or preparation of the manuscript"

"I have read the journal's policy and the authors of this manuscript have the following competing interests"

6. Please ensure that you refer to Figures 1 to 6 in your text as, if accepted, production will need this reference to link the reader to the figure.

Reviewers' comments:

Reviewer's Responses to Questions

**Comments to the Author**

1. Is the manuscript technically sound, and do the data support the conclusions?

Reviewer #1: Partly

2. Has the statistical analysis been performed appropriately and rigorously? 

Reviewer #1: Yes

3. Have the authors made all data underlying the findings in their manuscript fully available?

Reviewer #1: No

4. Is the manuscript presented in an intelligible fashion and written in standard English?

Reviewer #1: Yes

5. Review Comments to the Author

Reviewer #1: The research article entitled "Haematological predictors of poor outcome 1 among COVID-19 patients admitted

2 to an intensive care unit of a tertiary hospital in South Africa" by Chapanduka et al., has sough to identify the hematological parameters that can be used as prognostic biomarker for African ethnicity.

The study is very interesting and it is well written yet the introduction and the discussion need to be more succinct.

Tha major limitation of this study is the small number of patients that have been used. Larger cohort could have been better to find significant correlation with the same of the markers that have already been defined in other studies.

There are few concerns that the authors may need to address:

1. Have the authors measured the level of serum ferritin? if yes, how was it correlated with the disease severity?

2. The authors have shown the ROC for CRP with the data not to be found anywhere. Can you please include these data in your manuscript?

3. Have the authors checked the Univariate and multivariate analysis of the identified risk factors?

4. Generating a table with receiver operating characteristic analysis of promising markers for severity of COVID-19 may better clarify the article conclusion.

6. PLOS authors have the option to publish the peer review history of their article (what does this mean?). If published, this will include your full peer review and any attached files.

Reviewer #1: No

---

## [Author Response · Author response to Decision Letter 0]

15 Mar 2022

Reviewers’ responses to reviewers

Response to Reviewer 1 

Thank you for your review of our paper and for the valuable recommendations. We tried to make the introduction and discussion more concise. Changes are made in lines 94-96, 114-119, 131-132, 282-284, 292-295, 305 (manuscript with track changes). We have answered each of your points below.

1. Have the authors measured the level of serum ferritin? if yes, how was it correlated with the disease severity?

Response: The serum ferritin was not measured in this study

2. The authors have shown the ROC for CRP with the data not to be found anywhere. Can you please include these data in your manuscript?

Response: Thank you for highlighting this typing error. The ROC figure with CRP has been removed. 

3. Have the authors checked the Univariate and multivariate analysis of the identified risk factors? 

Response: Yes, both univariate and multivariate analysis to identify risk factors were performed. Table 3 shows univariate (unadjusted) and multivariate (adjusted) analysis of risk factors. 

4. Generating a table with receiver operating characteristic analysis of promising markers for severity of COVID-19 may better clarify the article conclusion. 

Response: We have provided a table of the ROC characteristic of the promising severity markers of COVID-19 (Table 4)

---

## [Decision Letter · Decision Letter 1]

18 Jul 2022

PONE-D-21-33196R1Hematological predictors of poor outcome among COVID-19 patients admitted to an intensive care unit of a tertiary hospital in South AfricaPLOS ONE

Dear Dr. Nyasulu,

Thank you for submitting your manuscript to PLOS ONE. After careful consideration, we feel that it has merit but does not fully meet PLOS ONE’s publication criteria as it currently stands. Therefore, we invite you to submit a revised version of the manuscript that addresses the points raised during the review process.

The reviewers have minor comments this time. Please address their concerns prior to resubmission==============================

We look forward to receiving your revised manuscript.

Kind regards,

Monica Cartelle Gestal, PhD

Academic Editor

PLOS ONE

Journal Requirements:

Reviewers' comments:

Reviewer's Responses to Questions

**Comments to the Author**

1. If the authors have adequately addressed your comments raised in a previous round of review and you feel that this manuscript is now acceptable for publication, you may indicate that here to bypass the “Comments to the Author” section, enter your conflict of interest statement in the “Confidential to Editor” section, and submit your "Accept" recommendation.

Reviewer #2: All comments have been addressed

Reviewer #3: (No Response)

2. Is the manuscript technically sound, and do the data support the conclusions?

Reviewer #2: Yes

Reviewer #3: Yes

3. Has the statistical analysis been performed appropriately and rigorously? 

Reviewer #2: Yes

Reviewer #3: Yes

4. Have the authors made all data underlying the findings in their manuscript fully available?

Reviewer #2: Yes

Reviewer #3: Yes

5. Is the manuscript presented in an intelligible fashion and written in standard English?

Reviewer #2: Yes

Reviewer #3: Yes

6. Review Comments to the Author

Reviewer #2: The manuscript by Chapanduka et al, addresses in the critical-care setting the role of blood tests to predict poor outcomes. Their finding concur with previous observations in COVID-19. The study is well-designed and the analysis adequate. The manuscript is well-written. As a minor suggestion, a few early (and relevant) studies were not included or discussed where similar findings had already been reported (e.g., PMID: 32503668, PMID: 33549831, PMID: 32283162) and a meta-analyses used instead (Ref #10). These should be incorporated and discussed in context, as the populations in those manuscripts had similarities and differences with the cohort presented in this manuscript.

Reviewer #3: The finding that was most significantly different from the mainstream results was the higher rate of female mortality, yet they do not address potential explanations for this. Could this be associated to other confounding external factors like obesity, or other demographic factors, pre-existing comorbidities or acute ones developed during the ICU admission?

7. PLOS authors have the option to publish the peer review history of their article (what does this mean?). If published, this will include your full peer review and any attached files.

Reviewer #2: No

Reviewer #3: **Yes: **Juanita Valdes Camacho, M.D

---

## [Author Response · Author response to Decision Letter 1]

17 Aug 2022

Response to Reviewer #2 

Thank you for your review of our paper. We hereby give a point-by-point response to the comments provided. 

Comment 1: The manuscript by Chapanduka et al, addresses in the critical care setting the role of blood tests to predict poor outcomes. Their finding concurs with previous observations in COVID-19. The study is well-designed and the analysis adequate. The manuscript is well-written. As a minor suggestion, a few early (and relevant) studies were not included or discussed where similar findings had already been reported (e.g., PMID: 32503668, PMID: 33549831, PMID: 32283162) and a meta-analyze was used instead (Ref #10). These should be incorporated and discussed in context, as the populations in those manuscripts had similarities and differences with the cohort presented in this manuscript

Response 1: Thank you very much, we highly value your recommendation. In the tracking manuscript line 304-326, The articles suggested were added and discussed in the manuscript as follows: “Liu et al., 2020 investigated the baseline NLR of 245 patients admitted to hospital with COVID-19 and found 8% higher risk of in-hospital mortality for each unit increase in NLR and 15-fold higher risk of death in patients with high NLR tertile [30]. However, in addition to high NLR, these patients were found to be older, more likely to be males, and have underlying comorbidities, compared with patients in the lowest tertile of NLR [30]. Núñez et al., 2020 investigated baseline characteristics of 282 hospitalised patients with COVID-19 and demonstrated that high NLR, at the moment of hospitalisation, predicts poor outcomes including death or critical care admission [31]. In contrast to our findings, they reported lymphopenia is a good independent predictor for ICU admission, mechanical ventilation, and death [31]. While the increased NLR was mostly due to the increase in neutrophils and lymphopenia was a constant finding in our ICU cohort, regardless of the outcome. Unlike our study which focuses on ICU patients with severe COVID-19, previous studies investigated the baseline NLR for patients at hospitalisation with variable COVID-19 severity [30,31]. On the other hand, Ma et al., 2020, investigated baseline characteristics for 81 patients with severe covid-19 and found a high NLR have higher incidence of ARDS and a higher rate of mechanical ventilation [32]. Comparably, analysis of leucocyte and neutrophil counts and the NLR in our ICU cohort, confirms the superiority of the NLR in the severe COVID-19 setting”.

Response to Reviewer #3 

Thank you for your review of our paper. Below is a point-by-point responses to the comments provided 

Comment 1: The finding that was most significantly different from the mainstream results was the higher rate of female mortality, yet they do not address potential explanations for this. Could this be associated to other confounding external factors like obesity, or other demographic factors, pre-existing comorbidities or acute ones developed during the ICU admission?

Response 2: Thank you very much for the valuable comment. Interestingly, there were no confounding factors contributing to the higher mortality based on the available data. While this study focused on common risk factors of severe COVID-19 such as age, smoking status, and common comorbidities, other factors such as ethnicity and host genetic predisposition were not explored in this study. This study highlights the need for studies that focus on host genetic predisposition in African population. 

In the revised manuscript, statistical analysis added to exclude confounding factor (lines 225-231 and Appendix-1,2) which states “Further analysis of the data, to elucidate the association of gender and comorbidities with mortality did not reveal any association between mortality and comorbid conditions namely, obesity, hypertension, diabetes mellitus and acute kidney injury (Appendix 1). Age was significantly higher among non-survival females (p<0.001) and further analysis was carried out to identify any gender differences between the various age categories. This analysis demonstrated no significant gender difference between the various age categories, as summarised in Appendix 2., and lines 408-417 paragraph has been added to the discussion which states: “In contrast to previous studies that demonstrated higher mortality among males, our study revealed a higher mortality among female patients [44, 45]. The description of the study sample by sex was similar. We conducted stratified analysis to assess the role of age and comorbidities on increased risk of death among female patients. No association identified that could explain female gender being at higher risk of mortality among COVID-19 patients admitted in the ICU. The comorbidities specifically analysed were obesity, hypertension, diabetes, and acute kidney injury. Advanced age had a statistically significant correlation with the risk of mortality in the study population in general regardless of gender (Table-1, Appendix-2). Considering that the Western Cape Province of South Africa has a unique ethnic profile, other possible risk factors such as host genetic predisposition need to be explored further [46].”

---

## [Decision Letter · Decision Letter 2]

26 Sep 2022

Hematological predictors of poor outcome among COVID-19 patients admitted to an intensive care unit of a tertiary hospital in South Africa

PONE-D-21-33196R2

Dear Dr. Nyasulu,

We’re pleased to inform you that your manuscript has been judged scientifically suitable for publication and will be formally accepted for publication once it meets all outstanding technical requirements.

Kind regards,

Monica Cartelle Gestal, PhD

Academic Editor

PLOS ONE

Additional Editor Comments (optional):

Reviewers' comments:

Reviewer's Responses to Questions

**Comments to the Author**

1. If the authors have adequately addressed your comments raised in a previous round of review and you feel that this manuscript is now acceptable for publication, you may indicate that here to bypass the “Comments to the Author” section, enter your conflict of interest statement in the “Confidential to Editor” section, and submit your "Accept" recommendation.

Reviewer #2: All comments have been addressed

2. Is the manuscript technically sound, and do the data support the conclusions?

Reviewer #2: Yes

3. Has the statistical analysis been performed appropriately and rigorously? 

Reviewer #2: Yes

4. Have the authors made all data underlying the findings in their manuscript fully available?

Reviewer #2: Yes

5. Is the manuscript presented in an intelligible fashion and written in standard English?

Reviewer #2: Yes

6. Review Comments to the Author

Reviewer #2: All my concerns have been addressed, as reflected in the manuscript. In my view, the manuscript is ready for publication.

7. PLOS authors have the option to publish the peer review history of their article (what does this mean?). If published, this will include your full peer review and any attached files.

Reviewer #2: No
